# Reaching adolescents with health services: Systematic development of an adolescent health check-ups and wellbeing programme in Ghana (Y-Check, Ghana)

Benedict Weobong [1,2] *, Franklin N. Glozah [1], Hannah B. Taylor-Abdulai [3], Eric Koka [4], Nancy Addae [4], Stanley Alor [1], Kid Kohl [5], Prerna Banati [6], Philip B. Adongo [1], David A. Ross [7]

1 School of Global Health, Faculty of Health, York University, Toronto, Canada, 2 Department of Social and Behavioural Sciences, School of Public Health, University of Ghana, Accra, Ghana, 3 Department of Physician Assistant Studies, University of Cape Coast, Cape Coast, Ghana, 4 Department of Sociology and Anthropology, University of Cape Coast, Cape Coast, Ghana, 5 Technical Advice and Partnerships Department, The Global Fund, Geneva, Switzerland, 6 Adolescent and Young Adult Health Unit, Maternal, Newborn, Child and Adolescent Health and Ageing Department, World Health Organization, Geneva, Switzerland, 7 Institute for Lifecourse Health Research, Stellenbosch University, Stellenbosch, South Africa

* bweobong@yorku.ca

**Data Availability Statement:** All relevant data for this study are publicly available from the Qualitative

## Abstract

### Background

Routine health check-ups may improve adolescent health, but global guidelines are lacking. Phase 1 of the WHO-coordinated Y-Check Research Programme involved three African cities to co-produce a programme of adolescent health check-ups. We describe a systematic approach to developing a routine adolescent health check-ups and wellbeing programme (Y-Check) to contribute evidence on whether adolescent health check-ups should be part of routine health services in Ghana.

### Methods

Y-Check Phase 1 was conducted in four communities in Cape Coast Ghana, over two stages using a variety of methods: (a) needs assessment and landscape analysis on the health of adolescents (existing policies/programmes, school system, adolescent health conditions) was conducted through desk-review and interviews with key informants to identify the potential content, delivery strategy and settings for adolescent health check-ups in this context; (b) co-designing the Y-Check intervention framework through person-centred participatory workshops and a consensus-building workshop with multiple stakeholders, including adolescents (10–19 years) and their parents. The study was conducted between January 2020 and October 2020.

### Results

The Y-Check intervention consists of two check-ups with content that is tailored to the needs of younger adolescents and older adolescents; delivered at both school and

Data Research repository (https://doi.org/10.5064/F6ILXCJT).

**Funding:** This study received funding from Fondation Botnar (REG-19-007) through a Technical Service Agreement with World Health Organization as the primary recipients of the funding, however the views expressed do not necessarily reflect Fondation Botnar's official policies. The funders had no role in study design, data collection and analysis, decision to publish, or preparation of the manuscript.

**Competing interests:** The authors have declared that no competing interests exist.

community settings by a team of trained staff in multiple steps involving up to four stations. Y-Check includes a referral system for adolescents with any problems that cannot be investigated or treated on-the-spot.

## Conclusions

Our systematic approach to co-producing Y-Check has resulted in an intervention whose content and structure is determined by the local context, and which was adjudged by multiple stakeholders to be likely to be both useful and acceptable, and which builds on best practice. As a logical next step, the Y-Check will be subjected to pilot testing and implementation research to rigorously evaluate the feasibility, acceptability, coverage, yield of previously undiagnosed conditions and cost of these health check-ups.

## Introduction

Without doubt, adolescents today have a better chance of surviving in good health, yet an estimated 1.2 million still die each year–mostly from preventable causes [1], with the majority of these deaths occurring in low- or middle-income countries (LMICs) (ibid). Lack of access to essential information, quality services and protective environments have been implicated as important contributory factors to the remaining burden of adolescent poor health [2]. The importance of paying attention to the complex nature of health and wellbeing at this stage of life has been strongly argued [3], and this calls for bold, innovative, and equitable interventions.

Several interventions have been developed to provide for the health care needs of adolescents in high income countries, and effective, evidence-based interventions are available for countries to protect and promote the health of adolescents [1]. Further, WHO provides clear, evidence-based guidance on whether national public health systems should provide health screenings for a diverse set of conditions including cervical cancer, hearing or newborn screening but these are largely targeted to young children or older adults and such guidance is lacking for adolescents (10–19 years old) in LMICs. This is against the backdrop that many adolescents and young people in LMICs, including Ghana, grow through adolescence without receiving any preventive or promotive health services [4]. The few who do attend health services, often do so late due to multiple barriers, including a lack of money for transport or for health service payments, needing permission from an adult or reticence to disclose health concerns to health workers [5, 6]. This creates avoidable missed opportunities for early detection and treatment. Early detection and treatment of health problems can immediately benefit adolescents [4] and adopting healthy behaviours and appropriate health care-seeking during adolescence is likely to reduce morbidity, disability and premature mortality both during adolescence and later in adulthood [7].

Similar to other countries, Ghana has an adolescent health policy that supports a programme for Adolescent Health and Development (ADHD). The ADHD programme aims to provide comprehensive health services and accurate health information for individuals between the ages of 10 and 24 years. However, health services for adolescents are largely not integrated, are of poor and uneven quality and coverage, with inequity in access and utilization, are generally limited to sexual and reproductive health (SRH), HIV and sexually transmitted infections (STIs) and do not fully address the broader health and health-related

problems faced by adolescents [8]. There are no routine health check-ups for adolescents, though the Government through the Ministries of Education and Health introduced mandatory health screening for all first-year senior high school students in Ghana in 2017. Whilst this is a commendable initiative, its introduction and implementation has not been without challenges such as health and education systems issues, contrary cultural belief systems regarding the notion of health check-ups, and policy enforcement challenges. There are thus opportunities for the introduction and scale-up of routine health check-ups for adolescents in Ghana.

However, despite a strong theoretical case [9], empirical evidence is lacking on the effectiveness of adolescent health check-ups, the detailed content, feasibility, mode of delivery and most appropriate age(s) for such check-ups [10]. Research is needed to fill the evidence gap on the effectiveness and cost of adolescent health check-ups for WHO and partners to produce evidence-based guidance on whether national public health systems should provide routine adolescent health check-ups and, if so, at what ages, with what content, and through what delivery mechanisms. WHO has taken the initiative to fill this evidence gap by coordinating research in three African cities (Chitungwiza Zimbabwe, Mwanza Tanzania, Cape Coast Ghana). In this paper, we report the systematic development of a co-created routine health check-up intervention (Y-Check) for adolescents in Ghana. The development process as described was guided by the following objectives: 1) to determine what the content should be for the health check-ups; 2) to determine the best delivery strategy and setting for the health check-ups.

## Materials and methods

### Study design

The framework for co-production and prototyping set out by Hawkins et al [11], utilising a transdisciplinary participatory action research (TDAR) method, was employed to assess feasibility and intervention characteristics. The intervention development involved three stages (a) identifying key contextual information on adolescent health; (b) identifying specific components of the health check-up intervention that are feasible and acceptable to adolescents, parents, and other stakeholders; and (c) developing a theoretical framework for the health check-up intervention. The procedures in each stage were implemented as shown in Fig 1. We present the specific methods and results of each stage below to demonstrate how the health check-up intervention evolved through each of these stages of development. Broadly, we conducted a situation analysis which included both a needs assessment and a landscape analysis. A variety of methods were used including: literature review, elicitation of factual information provided through key informant interviews (KIIs) and through participatory workshops. Key informants comprised a cross-section of stakeholders including: key staff of Ghana Health Service; Ghana Education Service; leaders of relevant NGOs and community-based organizations working on adolescent health; younger adolescents, older adolescents, and parents of adolescents. Technical assistance was provided by researchers from WHO. Ethical clearance was obtained from the Ghana Health Service Ethics Review Committee after which the WHO Ethics Review Committee also gave their approval for the study. Written informed consent and written informed assent were obtained from parents/guardians and younger adolescents (<18 years) respectively prior to primary data collection. Data collection for the study took place between January 2020 and October 2020. Prior to this, we undertook preparatory works for the study which included: obtaining ethical clearance, identifying and recruiting study sites, and engaging key stakeholders as part of our community entry and engagement. The

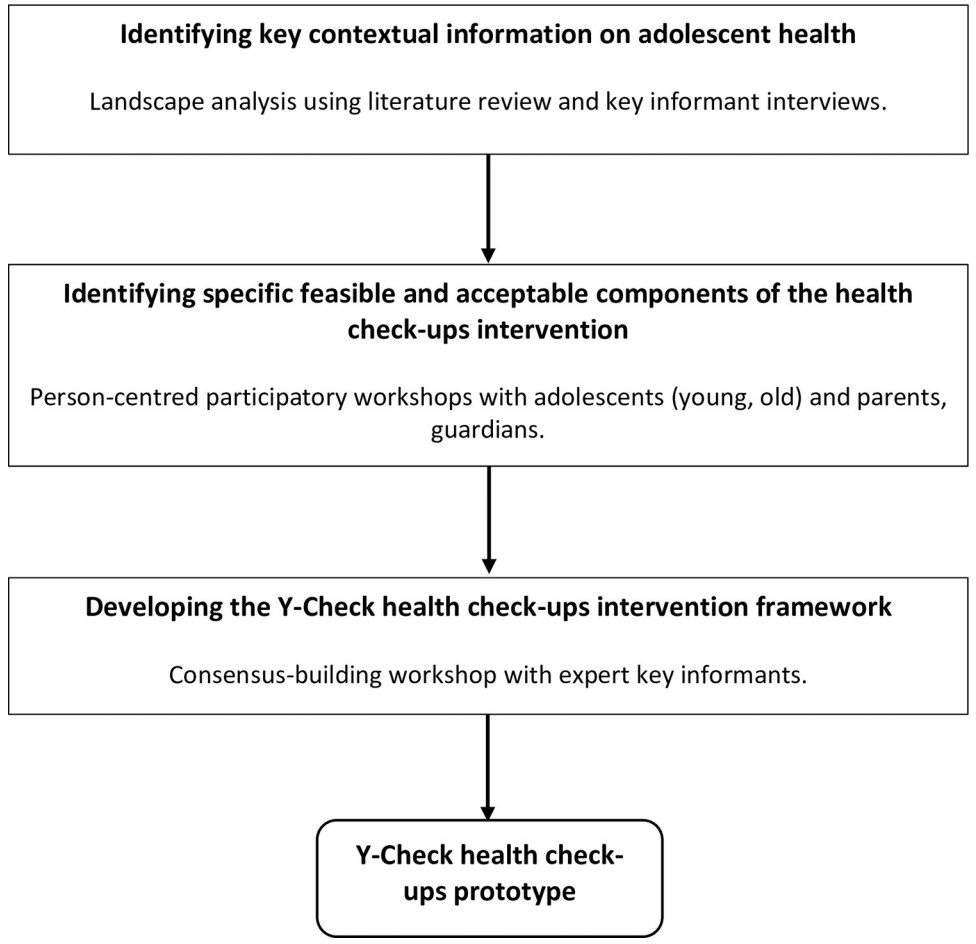

**Fig 1. Overview of the development process of Y-Check routine check-ups.**

preparatory works also included replanning to accommodate the challenges brought on by COVID-19. All these activities did not involve primary data collection for the study.

## Study setting

The study was carried out in the Cape Coast metropolitan area of the Central Region of Ghana, led by researchers from the School of Public Health, University of Ghana, and University of Cape Coast. The study was conducted in four communities (Abura, Efutu, Akon, and Kwaprow) within the Cape Coast metropolitan area which is in the coastal belt of Ghana. Abura and Kwaprow are relatively affluent communities with trading being the main source of livelihood. Akon and Effutu are relatively poorer communities where fishing and farming dominate economic activity, respectively.

## Stage 1: Identifying key contextual information on adolescent health

The goal of this stage was to obtain information on several relevant factors including, the conditions to screen for among adolescents, the evidence-based treatment or counselling for adolescents, school attendance, and current adolescent health policies and screening programmes, drawn from evidence from both national and local studies. This stage involved three steps: identification of key adolescent health policies, programmes, and interventions (Step A);

understanding the school health system and school attendance (Step B); and collection of estimates of the prevalence and severity of adolescent health problems by age and sex (Step C).

**Step A: Identification of key policies, programmes, and interventions for adolescent health.** Our formative research began with a detailed landscape analysis of key policies, programmes, and interventions on adolescent health, using literature reviews and key informant interviews. Key informant interviews (Table A in S1 Appendix), using a semi-structured interview guide, were conducted with key staff of the Ghana Health Service (GHS) and Ghana Education Service (GES), leadership of pro-adolescent non-governmental organizations and community-based organizations in the study area.

We conducted a rapid literature review. The main reason why we chose this methodology was to narrow the scope of the research question and reduce the extent of data abstraction with the aim to provide evidence in a timely and cost-effective manner. Rapid reviews are particularly useful for providing timely information for decision-making, especially when resources to conduct a full systematic review are [12] limited. Our search strategy involved a comprehensive search of electronic databases (PubMed, Embase, PsycINFO and Scopus; and Websites of Government of Ghana, relevant Ministries, Departments and Agencies) was done to identify relevant studies on adolescent health policy and interventions conducted in or on Ghana over the previous 5 years. Both peer-reviewed articles and grey literature were included in this review regardless of the study design.

*Inclusion and exclusion criteria.* All published peer-reviewed articles and grey literature papers on adolescent health policy and interventions in the 5-year period (2015 to 2020) were included in this review.

*Key search terms.* The search strategy for electronic databases incorporated both medical subject headings (MeSH terms) and free-text key terms adapted to suit each individual database using applicable controlled vocabulary. The key search terms used in this review are (("adolescent health"[All Fields] OR ("youth health"[MeSH Terms] OR ("young people health"[All Fields] AND "policies"[All Fields]) OR "programmes"[All Fields])) AND ("ghana"[MeSH Terms] OR "ghana"[All Fields]).

**Step B: Understanding the school health system and school attendance.** Two approaches were also employed in this step: (a) semi-structured interviews with key staff of GHS/GES related to teachers involved in the school health programme and related to the role of school nurses; and (b) a review of the published and unpublished literature on Ghana's school health programme, as well as basic and senior high school attendance. This was important to help determine the most appropriate setting (school, out-of-school/community) to deliver the health check-up intervention. Understanding school attendance coverage was particularly important for the study site given its peculiar secondary school infrastructure; Cape Coast has a reputation for having many of the best senior high schools in Ghana and thus attracts prospective students with the best entry qualification grades from all over the country. This means prospective students from the Cape Coast area may not have easy access to the local schools and thus potentially remain out of school or must attend schools elsewhere. Another important reason for collecting data on school attendance/access is because of the introduction of the free senior high school education policy in 2017. We wanted to know if this policy would have impacted on school attendance particularly for adolescents in the Cape Coast area based on the most recent enrolment year (2019/2020).

**Step C: Identification of common adolescent health problems.** In order to determine priority health conditions for the health check-ups, two approaches were employed in this final step in Stage 1: semi-structured interviews with key staff of GHS and leaders of relevant NGOs working on adolescent health; and literature review on the burden of adolescent health conditions in Ghana and the study area. The semi-structured interviews were designed to establish

opinions of respondents on what were the most common health conditions affecting adolescents, both nationally and in the study area. The interviews additionally looked for evidence on health conditions that are a priority for government interventions. The literature review looked for surveys of health conditions in adolescents in both published research articles and relevant government documents (we searched PubMed, Medline, PsycINFO, Websites of Government of Ghana, relevant Ministries, Departments and Agencies, Scopus, Google Scholar).

## Stage 2: Identifying specific components of the health check-ups intervention

The goal of this stage was to co-produce the health check-up intervention by working with younger (10–14 years) and older (15–19 years) adolescents, and their parents in the selection of feasible and acceptable content, delivery strategies, and settings. This stage begun with a series of person-centred participatory workshops with adolescents (young/old) and parents/guardians, using a multi-stage cluster sampling technique. First, a simple random sampling technique was used to select primary 1 to primary 6 and JHS 1 to JHS 3 students (approximately 6 to 14 years of age) in four basic schools (two private and two public from affluent and relatively poor socioeconomic backgrounds) within the Cape Coast Metropolitan area. Following this, four communities within the catchment area of the selected schools were selected using simple random sampling techniques. These communities provided access to out-of-school older adolescents. Within each of the selected schools, a simple random sampling technique was used to select students across five school years (primary 5 and 6 and JHS 1, 2 and 3), after stratification by sex and school year (2 participants per school year). For the out-of-school older adolescents within the selected communities, a central point was first identified using a map of the area, and a bottle was spun to randomly pick a direction of the first house. Subsequent houses were selected from this point using a fixed sampling interval of every second house in a straight line until reaching the boundary of the community. At the household level, simple random sampling was used to select participants. The next adolescent was approached if there was a refusal. A similar procedure was used in selecting parents/guardians of younger and older adolescents, separate from the adolescents recruited for the workshops. The workshops employed participatory learning and action (PLA) techniques such as free listing, ranking [13] and visualisation in participatory processes [14], to actively engage the participants and make the workshops exciting. For example, structured workshop guides were used to guide the completion of four key activities: (1) indicating yes or no regarding previous experience of participating in routine health check-ups; (2) indicating yes or no regarding acceptability of routine health check-ups; (3) ranking and free listing for preferred content (health areas), setting (school, community), method of delivery (health service staff, teachers) of health check-ups; and (4) using free listing to indicate barriers/facilitators for uptake of routine health check-ups. We had planned to recruit 20 participants for each group and conduct a series of three workshops each with the same participants of younger/older adolescents, but this was later revised to holding only one each with 12 participants per group due to COVID-19 restrictions.

## Stage 3: Developing the health check-ups intervention framework

The final stage in the intervention development process aimed at formulating a framework for the health check-up intervention. The goal was to reach a consensus on the number, content and method of delivery of the adolescent health check-ups. This involved a consensus-building workshop with all the key informants involved in stage 1 (experienced key staff of the Ministry of Health/Ghana Health Service, Ministry of Education/Ghana Education Service, relevant

NGOs and Community-Based Organisations, teachers and health workers). Ranking and scheduling methods were employed to finalise the list of content areas, delivery setting, and method of delivery of the health check-up intervention. For example, participants were presented with a schedule of discrete choices that the research team needed to make and each individual was asked to independently select one option for each of the choices. Following this, the choices were ranked.

## Data analysis

Framework analysis was used to analyse the qualitative data. Framework analysis involves the researchers who have been involved in the collection of the qualitative data and those who will be involved in the analysis, selecting and organising a set of codes into categories [15]. In the process of the framework analysis, we manually used the five stages of framework analysis outlined by Ritchie and Spencer [16]: familiarization, identifying a framework, indexing, charting, and mapping and interpretation. These phases enable us to understand and interpret data, and move from descriptive accounts to a conceptual explanation of what is happening from the qualitative data collected from participants in the study. These five stages were applied specifically to analyse data obtained through key information interviews, participatory workshops, and consensus-building workshop. Feasibility and acceptability were qualitatively assessed using two main data sources: the key informant interviews with health services/education staff and adolescents and their parents. The final determination of these critical implementation outcomes was arrived at during the consensus building workshop. To ensure the reliability and validity of the findings, two researchers first read and familiarized themselves with the data. Following this, few randomly selected interviews were read and initial codes generated through coding parcels of data in a systematic fashion. Based on the coded data and the original research questions, the researchers defined and collated codes into potential themes and finalized the codebook. The two researchers then coded the entire data set. Inter-rater reliability was tested by double coding ten randomly selected interview transcripts. For the rapid literature review, data was extracted using a standardized template. The data was synthesised using thematic analysis and reported as either ranks, prioritisation, estimation, or observation of trends. The primary (interviews and workshops) and secondary (document review) data were both used to (1) identify key policies, programmes, and interventions for adolescent health, (2) assess the school health system and school attendance, (3) identify common adolescent health problems, (4) identify specific components of the health check-ups intervention, and (5) determine the health check-ups intervention framework.

## Results

### Participant characteristics

A total of 172 participants were involved in the study comprising of 16 Key Informants (10 male); 41 younger adolescents (21 female) with a mean age of 12 years, and their parents; and 37 older adolescents (22 female), with a mean age of 16 years, and their parents.

### Stage 1: Key contextual information identified

**Key policies, programmes, and interventions for adolescent health.** There is strong policy and programmatic support for adolescent health programming in Ghana, though there are gaps regarding stakeholder awareness of and access to relevant information regarding these policies. The Adolescent Health Services Policy and Strategy is the current policy document on adolescent health services in Ghana [8]. This sought to improve the previous National

Adolescent Health Development Programme (AHDP) strategic plan (2009–2015), as this was limited in scope with a focus on SRH, HIV and sexually transmitted infections. The current adolescent health policy thus reflects some of Ghana's new health commitments and aligns with the Sustainable Development Goals (SDGs), aiming to provide the enabling environment for quality health service delivery for adolescents and for achieving universal health coverage (UHC) for all ages as enshrined in SDG 3.8. It also considers the Global Strategy for Women's, Children's and Adolescents' Health (2015–2030) which is aligned to the SDGs. Other adjunct adolescent health policies include the Ministry of Gender Children and Social Protection (MoGCSP) five-year (2018 to 2022) strategic plan to address adolescent pregnancy. Ghana also has a supportive school health policy, and only recently in 2017 the Ghana Health Service and Ghana Education Service introduced mandatory health screening for year 1 SHS students. This is an important adolescent health policy that must be supported to ensure it achieves the desired impact. There are some concerns with the programme design and implementation described below, and Y-Check offers opportunities to collaborate and strengthen the existing programme.

*School health programme.* Ghana's school health programme is led by the Ghana Education Service (GES) with technical assistance from the Ghana Health Service (GHS). The government's flagship School Health Education Programme (SHEP) is led by the GES with the goal of ensuring the provision of comprehensive health and nutrition education and related support services in both public and private schools in Ghana. SHEP operates at four levels: national secretariat led by a national SHEP coordinator; regional offices led by regional coordinators; district/municipal/metropolitan level led by coordinators; and school level led by a school-based health coordinator (a teacher with additional specific responsibilities for leading the planning and implementation of School Health Education Programme (SHEP) activities). The GES SHEP unit collaborates closely with the GHS to provide periodic sensitization of students on health issues such as water and sanitation, personal hygiene, sexual and reproductive health in all basic schools and SHS across Ghana. The metropolitan SHEP coordinator at the district level, together with the school-based health education coordinator (SHEP Teacher), work together to provide these services in basic (primary and JHS) schools, where there are no dedicated school nurses. The SHEP teachers in these basic schools, in addition to being involved in preventive and health promotive work, provide first aid and referral to the nearest health facilities. The GHS is a key player in the school health programme at the SHS level. The GHS specifically provides health services to students in SHSs through school infirmaries/sick bays with a team involving a school-based health coordinator, student infirmarian (a student volunteer, usually in the senior years, with responsibility of managing the school infirmary/sick bay), and a school nurse. School nurses are qualified professional nurses with special training on school health and are employed by the GHS, and have prescribing rights. In the Cape Coast area, nine out of the 10 SHSs have functional infirmaries, including relatively more advanced clinic facilities (with visiting doctors for out-patient consultations) in two of these nine schools. Each of the nine SHS has at least one dedicated school nurse who treats common minor ailments and offer first aid treatment. School nurses posted by the GHS to the senior high school work from 8am to 4pm on weekdays only, so some of the SHSs which have boarding facilities arrange for additional nurses, who are paid by the Parent-Teacher Association. Also, every school is linked to a near-by health facility such as the University Health Services and the Metropolitan Hospital where severe cases are referred. In addition, school nurses have a responsibility to encourage the setting up of adolescent health clubs in the schools and provide health education and promote good health among students.

*Health screening.* As described in the previous section, the GES and GHS collaborate in administering the school health programme. This includes health screening at schools. A

notable example of this is the mandatory screening for all first year SHS students to ascertain their health history and status, which was introduced in 2017 as one of the packages under the Government's free Senior High School Education policy. The screening is free of charge to the student, and the content includes: medical history and general physical examination (including vital signs), eye test for visual acuity, hearing assessment–physical examination and screening questionnaire, oral health screening, STI screening using a screening questionnaire, Tuberculosis (TB) screening, nutrition assessment including Body Mass Index (BMI), laboratory examination (full blood count, test for sickle cell trait status, Sexually Transmitted Infections-STIs (gonorrhoea, syphilis), HIV, hepatitis B, and urinalysis). These content areas were informed by the work of a technical committee using data from a situation analysis conducted in 2013. Both government and private clinical laboratories are contracted to conduct the laboratory tests. The screening is conducted over a period of two to three days in each school by a team of 6 to 10 health workers, in multiple steps involving up to six stations. The screening results are uploaded directly to an online server owned by the Ghana Health Service. We were unable to identify a routine system for students and/or their parents to be informed of their screening results unless they are referred for treatment. Referrals are made by the screening team either to the school nurse (if minor and manageable) or externally to linked health facilities if adverse results (mostly laboratory-based) are detected.

*Adolescent-friendly spaces*. Currently the Cape Coast metropolitan area has no functional adolescent health corner in any of its government health facilities. Adolescent-friendly spaces are places where young people can go to, ask questions about their health (mostly sexual health), receive referral services, and have convenient visiting hours, privacy, and confidentiality [8]. This will encourage young people to be open-minded and ask all the questions that are bothering them without feeling intimidated or afraid and will encourage them to talk about their struggles. Generally, patronage of adolescent health corners is poor in Ghana and non-existent in the study area [8].

**Other service providers.** Other providers of adolescent health services in the Cape Coast area include local NGOs such as the Planned Parenthood Association of Ghana (PPAG). The PPAG is the only NGO in the study area that provides essential services in two key areas: promotion of positive sexual behaviours, and promotion of better health and nutrition, including referrals. Between 2016 and 2017, an individual private physician offered health screening to students in one SHS in Cape Coast. The screening included a physical examination (eye, ear, throat, lungs and heart examination for murmurs), plus standard blood tests for anaemia and urine tests for urinary tract infections. The laboratory tests were carried out first and the results of the laboratory tests were shared with the physician before he made a visit to the school to conduct the physical examinations. Prescriptions for medication were provided on the spot for minor ailments and referrals were made for further examination/management where necessary. These were suspended because of disagreements between the parties on the fees charged, and because of the introduction of the free mandatory health screening policy.

**School health system and school attendance.** *School characteristics*. Ghana's school education system comprises nine years of basic school education (i.e. six years primary -ages 6 to 11 and three years of Junior High School (JHS)–ages 12 to 14); and three years of Senior High School (SHS)–ages 15–17. Table 1 shows the characteristics of schools in the Cape Coast Metropolis. There are 142 primary, 106 JHS and 10 SHS in the Cape Coast Metropolis (Table 1). There are both public and private schools at the primary and JHS levels, with the majority being public schools. There are no private SHSs in the Cape Coast Metropolis. Half the SHSs are co-educational, while the other half are either boys-only or girls-only schools, with all schools located in urban areas. The authors were not able to obtain data on the gender composition of basic schools (i.e., primary and JHS) in the Metropolis. With regard to 'Day' or

**Table 1. Characteristics of schools in the Cape Coast metropolitan assembly.**

| School characteristics | Primary | JHS | SHS | Total |
|---|---|---|---|---|
| **Type of school** | | | | |
| Public | 66 | 62 | 10 | 138 |
| Private registered | 59 | 39 | 0 | 98 |
| Private not registered | 17 | 5 | 0 | 22 |
| *Total* | **142** | **106** | **10** | **258** |
| **School organization** | | | | |
| Boys only | Not known | Not known | 3 | Not known |
| Girls only | Not known | Not known | 2 | Not known |
| Co-educational | Not known | Not known | 5 | Not known |
| *Total* | **142** | **106** | **10** | **258** |
| **Locality of school** | | | | |
| Urban | Not known | Not known | 10 | Not known |
| Rural | Not known | Not known | 0 | Not known |
| *Total* | **142** | **106** | **10** | **258** |
| **Boarding/Hostel Facilities** | | | | |
| Day only | Not known | Not known | 0 | Not known |
| Day with Hostel | Not known | Not known | 1 | Not known |
| Mainly Boarding | Not known | Not known | 4 | Not known |
| Boarding/Day & Hostel | Not known | Not known | 0 | Not known |
| Boarding & Day | Not known | Not known | 5 | Not known |
| *Total* | **142** | **106** | **10** | **258** |

Boarding facilities, the majority of the SHSs are mainly boarding or mixed boarding and day, but there is one school that is only day with hostel accommodation facilities.

*Enrolment (basic schools).* **Primar**. The 2019/2020 enrolment data from the Ministry of Education (Table 2) shows a small decline (9.3%) in numbers from 4931 in P1 (primary 1) to 4472 in P6 (primary 6). There is an almost equal number of boys (14,254) and girls (14,213), with a substantial proportion of students (42.6%) attending private primary schools.

**JHS.** Table 2 shows that in the 2019/2020 academic year approximately two-thirds of the 12,195 JHS students were registered in public schools (8,525 students) and the other third in private schools (3,670 students). There was no drop off in numbers between P6 (n = 4472) and JHS 1 (4,630). However, the number of adolescents registered in JHS declined by 22.5% from 4,630 in the first year (JHS 1—approx. age 12 years) to 3,586 in the third and final year of basic education (JHS 3). This decline in numbers was similar in boys (22.5%) and girls (22.6%) and there were slightly more girls than boys in all three years of JHS.

**SHS.** In the same academic year, there were a total of 22,858 students in SHS (13,542 males and 9,316 females). There was a substantial increase in the numbers in SHS 1 compared to JHS 3. Also, there were more males (59.2%) than females, so there were 145 boys for every 100 girls within the SHSs.

The government implemented free SHS education for all public schools in 2017, and this has resulted in a substantial increase in enrolment in SHS 1 within the Cape Coast Metropolis from 5,726 before the implementation of the policy in the 2016/2017 academic year (Table B in S1 Appendix) to 6,027 and 8,908 in the 2017/2018 and 2018/2019 academic years respectively (Tables C and D in S1 Appendix), representing a 35.7% increase in the 2018/2019 academic year. Whilst a dip in enrolment was recorded between the 2018/2019 and 2019/2020 academic years, the 2019/20 numbers were still higher than in the 2016/17 academic year. This

**Table 2. School attendance in public and private institutions in the Cape Coast metropolis for the most recent academic year for which data are available (2019/2020).**

| | Approximate Age | Public | | | Private | | | Combined | | |
|---|---|---|---|---|---|---|---|---|---|---|
| **2019–2020** | | **Male** | **Female** | **Sub-total** | **Male** | **Female** | **Sub-total** | **Male** | **Female** | **Total** |
| Primary 1 | 6 | 1,257 | 1,174 | 2,431 | 1,253 | 1,247 | 2,500 | 2,510 | 2,421 | 4,931 |
| Primary 2 | 7 | 1,204 | 1,241 | 2,445 | 1,147 | 1,161 | 2,308 | 2,351 | 2,402 | 4,753 |
| Primary 3 | 8 | 1,359 | 1,380 | 2,739 | 1,070 | 1,010 | 2,080 | 2,429 | 2,390 | 4,819 |
| Primary 4 | 9 | 1,475 | 1,455 | 2,930 | 971 | 907 | 1,878 | 2,446 | 2,362 | 4,808 |
| Primary 5 | 10 | 1,426 | 1,495 | 2,921 | 881 | 882 | 1,763 | 2,307 | 2,377 | 4,684 |
| Primary 6 | 11 | 1,391 | 1,486 | 2,877 | 820 | 775 | 1,595 | 2,211 | 2,261 | 4,472 |
| **Total Primary** | | **8,112** | **8,231** | **16,343** | **6,142** | **5,982** | **12,124** | **14,254** | **14,213** | **28,467** |
| [a]JHS 1 | 12 | 1,526 | 1,675 | 3,201 | 682 | 747 | 1,429 | 2,208 | 2,422 | 4,630 |
| JHS 2 | 13 | 1,400 | 1,463 | 2,863 | 551 | 565 | 1,116 | 1,951 | 2,028 | 3,979 |
| JHS 3 | 14 | 1,154 | 1,307 | 2,461 | 558 | 567 | 1,125 | 1,712 | 1,874 | 3,586 |
| **Total JHS** | | **4,080** | **4,445** | **8,525** | **1,791** | **1,879** | **3,670** | **5,871** | **6,324** | **12,195** |
| [b]SHS 1 | 15 | 4,722 | 3,004 | 7,726 | - | - | - | 4,722 | 3,004 | 7,726 |
| SHS 2 | 16 | 5,628 | 3,683 | 9,311 | - | - | - | 5,628 | 3,683 | 9,311 |
| SHS 3 | 17 | 3,192 | 2,629 | 5,821 | - | - | - | 3,192 | 2,629 | 5,821 |
| **Total SHS** | | **13,542** | **9,316** | **22,858** | - | - | - | **13,542** | **9,316** | **22,858** |
| **Overall total** | | **25,734** | **21,992** | **47,726** | **7,933** | **7,861** | **15,794** | **33,667** | **29,853** | **63,520** |

[a]JHS: Junior High School

[b]SHS: Senior High School

increasing trend coupled with possible dropouts, may have resulted in there having been 22% more students in SHS 1 (approx. age 16 years; n = 7,726) than in SHS 3 (n = 5,821) in the 2019/2020 academic year. In the same academic year however, the largest number of students in SHS was recorded in SHS 2 with 9,311students. Cape Coast is particularly unusual in terms of SHS education because it has some of the best SHSs in Ghana and thus attracts many (usually affluent) students from outside the Metropolitan area of Cape Coast. This phenomenon presents some challenges in gaining admission to these schools by students from Cape Coast. The transition rate for JHS to SHS in Cape Coast Metropolis is not reported. However, the regional rate is reported at 77.7% [17]. If this rate is applied to schools in Cape Coast for the most recent academic year of 2019/2020 (Table 2), 2646 of the 3405 JHS 3 students in the previous academic year (Table D in S1 Appendix) may have transitioned to SHS 1 in Cape Coast. The implication is that for the most recent academic year 5080 students (7726–2646) enrolled in SHS 1 in public schools in Cape Coast may have come from outside the metropolitan area of Cape Coast. This is entirely plausible because of the introduction of the computerized school placement system that places final year JHS students to their preferred public SHS across Ghana. Another plausible explanation is that the 22% who were estimated to have been unable to enrol into SHS 1 may have been from less affluent backgrounds if there is a bias in recruitment of more affluent students to the Cape Coast SHSs. This situation is likely to improve with the continuous implementation of the free SHS programme that specifically addresses this challenge with the introduction of a 30% equity policy to ensure less affluent JHS graduates can gain admission to elite schools (such as those in Cape Coast). This policy seems to be working as Ghana's education sector performance report shows in Ghana as a whole, less affluent JHS graduates' intake in elite SHSs has increased to almost 50% (from 10%) since the implementation of the policy in 2017 [17].

**Table 3. Top ten contributors to DALYs among adolescents aged 10–14 years in Ghana.**

| Cause | Overall DALYs | | Sex distribution: Rate/100,000 (% total DALYs) | |
|---|---|---|---|---|
| | Rate/100,000 | % Of total* | Female | Male |
| 1. Mental Disorders | 1114.5 | 10.4 | 1109.0 (11.2) | 1120.0 (9.7) |
| 2. Neurological Disorders | 871.5 | 8.1 | 1025.0 (10.3) | 723.0 (6.3) |
| 3. Malaria | 783.4 | 7.3 | 665.0 (6.7) | 898.0 (7.8) |
| 4. Enteric Infections | 663.0 | 6.2 | 618.0 (6.3) | 707.0 (6.2) |
| 5. Transport Injuries | 644.1 | 6.0 | 403.0 (4.1) | 877.0 (7.6) |
| 6. HIV/AIDS | 535.0 | 5.0 | 535.0 (5.5) | 535.0 (4.7) |
| 7. Skin Disease | 533.5 | 4.9 | 577.0 (5.8) | 491.0 (4.3) |
| 8. Unintentional Injuries (other than transport injuries) | 505.8 | 4.7 | 354.0 (3.6) | 652.0 (5.7) |
| 9. Neoplasms | 335.6 | 3.1 | 304.0 (3.1) | 367.0 (3.2) |
| 10. Musculoskeletal Disorders | 293.6 | 2.7 | 303.0 (3.1) | 285.0 (2.5) |
| 11. Other causes (grouped) | 4421.9 | 41.3 | 3977.5 (40.3) | 4850.6 (42.1) |

*may not add up to 100 because of rounding

**Common adolescent health conditions.** We observed that for all the secondary data obtained through literature search, the 2019 national Global Burden of Disease study (16) provided adolescent health conditions that was common to all the other literature and documents obtained. Based on the 2019 national Global Burden of Disease study, the top ten contributors to the burden of disease, which contribute 58.7% and 60% of the total burden, respectively, among younger adolescents (10 to 14 years) and older adolescents (15 to 19 years) in Ghana [18] are summarized in Tables 3 and 4. There are clear overlaps in the top ten contributors across both age groups, with mental disorders in the top two in both younger and older adolescents. Overall, the burden of disease is much higher among older adolescents compared to younger adolescents. The burden from injuries is particularly interesting with the number of Disability Adjusted Life Years (DALYs) lost per 100,000 population due to unintentional injuries other than transport injuries being more in younger adolescents compared to older adolescents. In both age groups, the burden from transport injuries is much higher among males. Skin diseases are also burdensome across both younger and older adolescents, but cancers are more of a burden in younger adolescents. Another important difference in the data is that cardiovascular diseases contribute more to DALYs among older adolescents but not younger

**Table 4. Top ten contributors to DALYs among adolescents aged 15–19 years in Ghana.**

| Cause | Overall DALYs | | Sex distribution: Rate (% total DALYs) | |
|---|---|---|---|---|
| | Rate/100,000 | % Of total | Female | Male |
| 1. Malaria | 1619.0 | 10.2 | 1598.3 (10.9) | 1639.5 (9.5) |
| 2. Mental Disorders | 1564.6 | 9.8 | 1665.8 (11.4) | 1464.2 (8.5) |
| 3. Transport Injuries | 1386.1 | 8.7 | 574.3 (3.9) | 2190.4 (12.7) |
| 4. HIV/AIDS | 1246.6 | 7.8 | 1276.5 (8.8) | 1216.9 (7.1) |
| 5. Neurological Disorders | 1151.7 | 7.2 | 1338.3 (9.1) | 966.8 (5.6) |
| 6. Skin Disease | 613.2 | 3.8 | 659.3 (4.5) | 567.5 (3.3) |
| 7. Enteric Infections | 579.5 | 3.6 | 540.8 (3.7) | 617.8 (3.6) |
| 8. Musculoskeletal Disorders | 516.1 | 3.2 | 536.7 (3.7) | 495.7 (2.9) |
| 9. Cardiovascular disease | 485.1 | 3.1 | 441.6 (3.0) | 528.1 (3.1) |
| 10. Unintentional Injuries (other than transport injuries) | 480.7 | 3.0 | 318.4 (2.2) | 641.6 (3.7) |
| 11. Other causes (grouped) | 6257.5 | 39.6 | 5627.8 (38.8) | 6881.9 (40.0) |

adolescents. Whilst maternal disorders are not in the top 10 contributors to DALYs even in older adolescents, the disaggregated data by sex suggest a relatively high burden in the older females (532.79/100,000 (3.66%)) so this cause would rank ninth in this age/sex group.

These align to a large extent with the target conditions identified in Ghana's Adolescent Health Service and Strategy document [8]. Also, within the study area in the Cape Coast Metropolis, data from the Health Directorate Information System for 2019 shows that the most frequently reported adolescent health problems among adolescents who sought health care were eye infections, anaemia, intestinal worms, sickle cell disease and injuries [19]. Furthermore, patterns emerging from Key Informant (KI) interview and workshops with adolescents and parents also show that eye sight, anaemia, psychosocial issues, growth and nutrition, oral health, and sexual health issues were common. Key informants also identified other conditions such as hearing impairment, and kidney problems, as health conditions of importance to adolescents in the study area. This background data on the common adolescent health conditions in Ghana and in the study area, along with consideration of whether there is a feasible, valid, low-cost screening tool for these conditions were important considerations in deciding the content of the intervention. Beyond this, particular attention was given to whether there is a cost-effective intervention (e.g. treatment, counselling, care, support) available that could be offered on-the-spot or through local referral.

## Stage 2: Potential health check-ups (Y-Check) components identified

**Feasibility and acceptability.**   To ensure community and stakeholder buy-in, a 16-member Community Advisory Board (CAB) that included young people as members, was successfully formed to advise the research team and provide community leadership and engagement for the project. The CAB consisted of the Director of Health Services in the Metropolis, representative of a Non-Governmental Organizations in Health (Planned Parenthood of Ghana), Metropolitan School-based Health Education Programme (SHEP) coordinator, two female and two male Head Teachers, three adolescents (two females and one male), two male local Assembly members, one female and one male Head of Health facilities, and two parents (one female and one male). The idea of introducing routine health check-ups for adolescents was endorsed in the interviews and consensus building workshop. The reasons for supporting this idea included the belief that adolescent health issues are a major concern in the Cape Coast metropolitan area that needed to be addressed through early detection. Participants also thought that a health check-ups programme would provide adolescents with the opportunity to discuss more openly about health issues of concern to them.

*Barriers and facilitators to uptake of routine health check-ups.* Notwithstanding the overwhelming community support and stakeholder buy-in, our study revealed important factors that could impede the uptake of routine health check-ups in the Cape Coast area unless the check-ups were designed appropriately. From the perspective of adolescents, the following were identified as facilitators or barriers for the uptake of the planned health check-ups: having the check-ups close and easily accessible; health system factors such as quality of services, cleanliness, and positive staff attitudes; incentives such as free health check-ups including treatment. To illustrate, the importance of positive staff attitudes was emphasized in the following quote in one of the workshops:

> "...want doctors who will understand us and our sensitive nature"-workshop with 10–14-year-old adolescents.

Others include: having health promotion/education messages and some entertainment at the location for the health check-ups; a reminder system to go for health check-ups; and the

**Table 5. Top ten health areas of concern to adolescents and parents.**

| Rank | Adolescents (10–14 years) | Adolescents (15–19 years) | Parents of Adolescents (10–14 years) | Parents of Adolescents (15–19 years) | Key Informants interviews |
|---|---|---|---|---|---|
| 1. | Home/family circumstances | Sexual Health | Home/family circumstances | Home/family circumstances | Eye screening |
| 2. | Growth and Nutrition | Eyesight | Eyesight | *Health Concerns | Teenage pregnancy |
| 3. | Eyesight | Home/family circumstances | Mental Health | Eyesight | Sexual Health Education |
| 4. | Anaemia | Mouth and Teeth | Sexual Health | Immunization | STIs |
| 5. | Deworming | Growth and Nutrition | Anaemia | Sexual Health | Dental Care |
| 6. | Sexual Health | *Health Concerns | Mouth and Teeth | Physical Disability | Hearing |
| 7. | Mouth and Teeth | Mental Health | Growth and Nutrition | Growth and Nutrition | Drug Addiction |
| 8. | Mental Health | Hearing | *Health concerns | Hearing | Mental Health |
| 9. | Physical disability | Physical Disability | Immunization | Deworming | Urine Routine Examination |
| 10. | Immunization | Anaemia | Physical Disability | Mental Health | Diabetes, Obesity |

*Health concerns: the opportunity to discuss other health issues of concern

STIs: Sexually Transmitted Infections

importance of promoting parental support for the check-ups. Some parents also stressed that the adolescent health check-ups should take place in a separate place from general health clinics for all ages, in order to maintain privacy and confidentiality. These opinions were largely shared by parents of adolescents. The following quote from the parent's workshop illustrates this point clearly:

> *"Adolescents who attend routine check-ups should not mingle with other patients. The team should create an adolescent section in health centers for only adolescents."*- workshop with parents.

From the perspective of KIs, the following were identified as important factors: the need to make the health check-ups free by integrating costs in the national health insurance scheme; making the experience of the health check-ups attractive such as engaging adolescents with information on health topics using tablet computers; having friendly staff; and upholding privacy/confidentiality to tackle the effects of stigma and poor trust in the health system.

**Content.** Through the participatory workshops and key informant interviews, participants identified key health areas that are of concern to adolescents and that they would want to be included in a health check-up programme. Table 5 shows the top 10 health conditions preferred by each group of respondents.

Across all respondent categories, issues of sexual health, eye health, mental health, and substance use were identified as the most preferred health areas for inclusion in health check-ups in the Cape Coast metropolitan area. Screening for oral health, physical disability, growth and nutrition status, and background assessment for potential stressors in the home environment of the adolescent, were the second most preferred areas for a health check-ups intervention.

**Delivery.** Preferences for mode and mechanics of delivery were explored. Adolescents were keen on having doctors or nurses to deliver the check-ups. The delivery should also provide opportunities for adolescents to play and learn on tablet computers during waiting time. The issue of attitudes also featured prominently as adolescents stressed that they should be talked to nicely by doctors/nurses during check-ups. In terms of costs associated with

accessing the check-ups, adolescents were almost unanimous that the service should be free and covered under Ghana's national health insurance scheme.

**Setting.** Participants discussed the most appropriate setting for health check-ups. Most of the key informants were of the opinion that the planned health check-ups programme should be delivered through school settings, for reasons of convenience and alignment with current approaches used in Ghana's school health education programme at both basic and senior secondary school levels. The idea of community settings also came up, albeit not as strongly as the preference for the school setting. Adolescents and parents were supportive of offering the check-ups in a community area that is easily accessible. For example, some participants thought the check-ups could also be provided at community centres because this provides the opportunity for adolescents who may be out of school for whatever reason to have access to the check-up and to also make the adolescents feel comfortable.

### Stage 3: Health and wellbeing check-ups (Y-Check) framework formulated

Fig 2 shows the evolving Y-Check framework. The framework identifies four key levels for the delivery of routine health check-ups for adolescents in Ghana: (1) the Health/Education Organisation (Ghana Health Service/Ghana Education Service) level; (2) dedicated Y-Check team

| | Awareness | Detection | Treatment | Referral | Enabling |
|---|---|---|---|---|---|
| **Health/Education Organisation (Ghana Health Service/Ghana Education Service)** | Engage and mobilise schools | | | | Promote positive staff attitudes |
| **Dedicated Y-CHECK team at Schools and/or communities** | | Age-appropriate tailored screening for target health areas | On-the-spot management | Sharing of results and quick referral | Use of voucher system |
| **Healthcare facilities** | | Conduct detailed assessment on referred cases | Provide treatment/interventions | Available services and accessible | Promote positive staff attitudes |
| **Digital health clubs** | Improve awareness and decrease stigma | | Provide peer support | Promote self-referrals | Provide peer support; reminders |

**Fig 2. Evolving Y-Check framework.** * to explore utility of this in phase 2. Figure adapted from PRIME mental health care plans framework [20].

at Schools and/or community level; (3) health facilities (including general primary care facilities, and specialist support) level; and (4) the digital health clubs level. Across these four levels, core activities will address five key domains of: raising awareness, support, and trust for the health check-ups; improving detection, providing treatment; ensuring timely referrals; and creating enabling environments. The next section discusses the emerging content, delivery plan, setting, and referral plan.

## Content

Based on the pooled data from the various stages of the systematic development process, given the disease burden of different health conditions (Tables 3 and 4) and the perceived importance of these by the various stakeholders (Table 5), tempered by cost and feasibility considerations, for Phase 2 we are proposing that all adolescents should be offered two check-ups, one when the adolescent is aged 10–14 years and one when they are aged 15–19 years. The plan is to offer both younger and older adolescents screening for the same set of health conditions, except screening for pre-hypertension and sexually transmitted infections/diseases (STIs/STDs) which would be limited to only older adolescents. As shown in Table 6, the content would include those health areas for which there are existing services in Ghana. Edutainment would be included as part of the content for adolescents as they wait to be screened.

**Delivery.**   The plan is to offer two health check-ups; the first check-up will be offered to young adolescents during early puberty and the second to older adolescents who are coming towards the end of puberty. A typical check-up session will entail multiple steps involving up to four stations or stages. Prior to each check-up session, key stakeholders, including the school leadership and the School Health Education Programmne (SHEP) coordinator will be consulted (at least two weeks ahead of the planned check-ups) to agree on a suitable date and time. Following this, on a typical check-up day, a team trained specifically on Y-Check comprising a young registration worker, medical laboratory scientist, and nurses/clinicians will set-up four separate but closely placed stations (either in the school or community area). Adolescents will be invited to the first station where they will be welcomed by the young registration worker who will carefully explain the process, administer the consent procedures, answer questions, create an electronic record form that can be accessed at all screening stations, and assist them to self-complete a digital screening questionnaire. In Station two, a nurse/clinician will conduct a physical examination—oral examination, hearing impairment, visual impairment, growth and malnutrition, and musculoskeletal tests. At station three the adolescent will be offered laboratory investigations by a medical laboratory scientist, testing for anaemia and STIs. At the fourth and final station a nurse/clinician will review the results from all the previous stations and provide treatment, counselling, advice, health-related information, a dental care pack, a menstrual pack, among others, and referral, as necessary.

## Setting

For younger adolescents, the check-ups will be offered to all students attending year 1 of Junior High School at a convenient area within the school. This setting will maximise yield (because of high attendance rate at this age band) and is convenient and strategic as the entry point for the first check-up. The second check-up will be offered at both the Senior High School year 1 (median age of 16 years- to strengthen the existing mandatory medical examinations) and community settings (e.g. community halls, CHPS-Community-based Health Planning and Services Compounds) for 16-year-old adolescents. This is because by this age, the proportion who are attending school has started to decline.

**Table 6. Health and well-being check-ups for younger and older adolescents, respectively.**

| Health condition or behaviour | Screening tool | Action | Age group | |
|---|---|---|---|---|
| | | | 10-14y | 15-19y |
| **Station 1. Registration and screening by electronic self-completion questionnaire on a tablet computer (Young research assistant)** | | | | |
| 1.1 Registration | Name, date of birth, etc | None | Y | Y |
| 1.2 Psychosocial screen | HEEADSSSS (**H**ome environment, **E**ducation and employment, **E**ating, **A**ctivities, **D**rugs, **S**exuality, **S**uicide/depression, **S**afety from injury and violence, **S**creen use, and **S**trengths)[1] | Further assessment by clinician, with on-the-spot treatment and/or referral if indicated | Y | Y |
| 1.3 Screening questions for physical impairments | Physical (Musculoskeletal) Screening Questions | Physical (Musculoskeletal) Tests | Y | Y |
| 1.4 Screening question for oral health | *Do you have any pain or swelling in your mouth today?* | Oral examination | Y | Y |
| 1.5 Screening question for convulsions | *Do you have convulsions, involuntary movement, rigidity or loss of consciousness?* | Further assessment by clinician, with on-the-spot treatment and/or referral if indicated | Y | Y |
| 1.6 Screening questions for common mental disorders (depression and anxiety) | Patients´ Health Questionnaire–Adolescent (PHQ-A); Generalised Anxiety Disorders Questionnaire (GAD) | Further assessment by clinician, with on-the-spot treatment and/or referral if indicated | Y | Y |
| 1.7 Screening for hazardous alcohol use and for tobacco use | Alcohol, Smoking and Substance Involvement Screening Test (ASSIST) | Further assessment by clinician, with on-the-spot treatment and/or referral if indicated | N | Y |
| 1.8 Male circumcision | *Are you circumcised?* | Physical examination (irrespective of answer). | Y (males only) | Y (males only) |
| 1.9 Immunization | Immunization record (when available) or verbal report related to HPV and DT vaccination during older childhood or adolescence (9y+) | Immunization with first dose by clinician if indicated, with referral for subsequent doses. | | |
| 1.10 Deworming (Soil-transmitted helminths) | *Have you received deworming tablets for soil-transmitted helminths within the past twelve months[a]?* | Provision of single-dose albendazole (400 mg) or mebendazole (500 mg) by the clinician if not received soil-transmitted helminth deworming within the past twelve months[a] | Y (high prevalence populations only) | N |
| 1.11 Schistosomiasis mass treatment | *Have you received mass treatment for schistosomiasis within the past x months[b]?* | Provision of single dose praziquantel. | Y (high prevalence populations only) | Y (high prevalence populations only) |
| **Station 2. Screening by visual inspection/clinical measurements part 1 (Community health worker or nurse, with specific additional training)** | | | | |
| 2.1 Oral health | Visual inspection for dental caries, pits or fissures in the enamel, abscesses, swelling or oral lesions | *No significant problems*: fluoride varnish, toothpaste and brush kit, session on oral health. *Dental caries and/or pits or fissures in the enamel*: As above + application of silver diamine flouride *Pain, swelling, abscess, oral lesions*: As above + referral to dentist | Y | Y |
| 2.2 Physical impairment | Tests for activities (n = 3), walking (n = 3), arm function (n = 3), if indicated by screening questions (1.3 above). | Further assessment by clinician, with referral to physiotherapist for further assessment and treatment if indicated | Y | Y |
| 2.3 Anthropometry | Mid-Upper Arm Circumference or Body Mass Index +/- Body Volume Index | Brief counselling and advice on diet and physical activity (all adolescents) Refer if severely underweight | Y | Y |
| 2.4 Male circumcision inspection | Genital inspection (males only) | If uncircumcised, advice from the clinician, and referral if requested | Y (males only) | Y (males only) |
| **Station 3. Screening by visual inspection/clinical measurements part 2 (Community health worker or nurse, with specific additional training)** | | | | |
| 3.1 Visual impairment | Portable Eye Examination Kit (PEEK)–Acuity for distance vision or Snellen´s E-Chart with own correction (if any). | Refer to optometrist. | Y | Y |
| 3.2 Hearing impairment | HearScreen or hearWHO smartphone app | Check for wax by clinician. If present, remove and repeat hearing test. If hearing still significantly reduced, refer to hearing specialist. | Y | Y |

*(Continued)*

**Table 6.** (Continued)

| Health condition or behaviour | Screening tool | Action | Age group | |
|---|---|---|---|---|
| | | | 10-14y | 15-19y |
| 3.3 Hypertension | Digital sphygmomanometer using the appropriate cuff size. 3 measurements after sitting quietly for ≥3 minutes. | Advice from clinician: *Suspected Prehypertension (Lowest systolic reading 120-129mmHg)*: Advice (diet, exercise, etc), advise repeat BP measurement after 6 months) *Suspected Hypertension (Lowest systolic reading ≥ 130mmHg)*: Advice (diet, exercise, etc), refer to specialist for diagnosis of cause. | N | Y |
| **Station 4. Laboratory tests (Laboratory technician)** | | | | |
| 4.1 Anaemia | Haemoglobin measurement (Hemocue) | Advice from clinician (5.1 below): *If anaemic (children <11 years Hgb <11.5g/dL, 12–14 years <12g/dl, females >15 years <12g/dl, males > 15 years <13 g/dL)*: Iron and folic acid-supplementation (3 months supply) *If severely anaemic (<8g/dl)*: Refer to physician for investigation | Y | Y |
| 4.2 STI | Xpert STI screen (NG, CT, TV) on urine | On-the-spot treatment by clinician | N | Y (in community venues only) |
| 4.3 HIV | HIV oral mucosal test (with confirmatory blood test if positive) | Initiation of treatment by clinician, with referral to HIV treatment clinic. | Y | Y (in community venues only) |
| Malaria (Malarious areas only) | Malaria rapid diagnostic test | *All adolescents*: Provision of free long-lasting insecticide-treated net *Malaria positive*: Clinician to treat for malaria. | Y | Y |
| **Station 5. Clinical consultation (Clinician or nurse practitioner)** | | | | |
| 5.1 Clinical consultation | Review of all results with additional assessment and/or physical examination, if indicated. The consultation will particularly focus on issues identified either in the electronic self-completion questionnaire (Station 1 above) or from any of the results from Stations 2–4 above) | Provision of on-the-spot treatment, care, or referral as indicated, based on the results from the check-up plus a brief medical consultation by the clinician | Y | Y |
| 5.2 Provision of health promotion advice and commodities | Not applicable | All participants will be offered: Health promotion literature Brief, individually-tailored counselling A free tube of toothpaste and toothbrush, plus a demonstration of how to brush teeth correctly All post-menarcheal girls will be offered: A free menstrual health kit All sexually active participants will be offered: Free male or female condoms (Where this is not permitted (eg. in schools in some countries), participants will be provided with information on where they can be obtained) | | |

**Key:**

(a) Soil-transmitted helminth (STH) mass treatment will only be offered in schools/communities where there is a pre-existing deworming programme in the study area in Ghana.

(b) Schistosomiasis mass treatment will only be offered in schools/communities where there is a pre-existing schistosomiasis mass treatment programme. The screening question will be tailored to the specifics of the local population using current WHO recommendations.

(c) Y = Yes, N = No

**Referral pathway.** The check-ups will include a referral component. This is a major weakness in current school health and other adolescent health programmes in Ghana. As outlined in the delivery plan, stage four of the check-up will involve a referral where necessary. The

Y-Check referral plan will involve first creating a database of credible and easily accessible service providers in the Cape Coast area who can offer high-quality, adolescent-friendly services. Following this, a memorandum of understanding will be reached with selected service providers, with preference given to accredited national health insurance facilities. For each referral, participants will need to provide informed consent and receive a referral "letter" which acts as a voucher to receive relevant services from the accredited provider. The provider will keep the "voucher" and this will allow the research team to know that the participant has actually attended for the referral in order for the electronic case record form to be updated. The voucher would also entitle the provider to payment for the services rendered to the participant–based on pre-agreed fees.

## Discussion

This paper reports the process taken in designing a routine health check-ups intervention for adolescents in Ghana. The paper also provides the intervention framework systematically developed from a person-centred approach. In summary, the proposed Y-Check routine health and wellbeing check-ups intervention in Ghana will consist of two check-ups with content that is tailored to the needs of younger adolescents (10–14 years) and older adolescents (15–19 years); delivered in schools only for the younger adolescents and at both school and community settings by a team of trained staff in multiple steps involving up to four stations or stages. A core component of Y-Check is a referral system with feedback to ensure the outcome of each referral is recorded.

The proposed Y-Check intervention is in step with Ghana's national policy and guidelines on the provision of adolescent health services. Based on the existing policy and guidelines, the planned study area in Cape Coast has three adolescent health services that are of relevance and interest to Y-Check: 1) the current GES-led mandatory screening for first year SHS students; 2) school health unit of the GHS; and 3) a private physician's medical screening model. Whilst these initiatives may have advanced the state of adolescent health services in the Cape Coast area, there are important concerns with implications for the design of routine health check-ups for adolescents. First, the current free screening in SHS does not appear to have a well-structured system for sharing and discussing the results with students and their parents for the necessary follow-ups, and there is no on-the-spot treatment for minor ailments. Furthermore, the content of the mandatory screening package does not include assessments for mental, neurological and substance use conditions. Mental health conditions account for a third of the burden of disease in adolescents globally [21], and more than half the burden of adult mental disorders has its onset in adolescence [22]. In Ghana, about one third of adolescents report at least moderate depressive symptoms on the PHQ-9 [23], and these symptoms tend to worsen over the course of adolescence. Risky sexual behaviour [24], substance use (cigarettes, alcohol, cannabis, and amphetamines) [25–27], and suicidal behaviour [28] are also significant problems during this period and often continue into adulthood. As noted by Glasner and colleagues these behavioural conditions can benefit from early detection through previsit psychosocial screening measures [29] such as the HEEADSSS standing for Home environment, Education and employment, Eating, Activities, Drugs, Sexuality, Suicide/depression, and Safety from injury and violence [30].

Four key lessons emerged from the development of Y-Check. Firstly, our study confirmed the existing literature on enablers and barriers regarding use of health services in general by adolescents. However, regarding the specific area of health check-ups, this was an entirely new concept to the adolescents and their feedback in the interviews confirmed this. Most adolescents and their parents had not considered visiting a health worker or clinic unless the

individual had already identified that they had a health problem (eg. pain or an injury). Important implementation considerations included: 1) Some parents and adolescents thought the frequency of health check-ups should be short i.e. either monthly or half yearly. The research team thought this may reflect difficulties in appreciating the feasibility and costs of health check-ups; 2) Some parents particularly wanted the check-ups to include screening adolescent girls for breast cancer despite the fact that breast cancer is exceptionally rare in this age group; and 3) Both parents and adolescents thought that the check-ups should be free of cost to them.

Secondly, this study highlights the importance of co-production in developing interventions for the promotion of adolescent health. The involvement of key stakeholders in the co-production of intervention content provides a mechanism for tailoring intervention content to the context and target population to maximise acceptability and reduce the likelihood of problems with implementation [11]. Co-production is also emerging as a useful methodology for developing school-based health interventions [31]. Adolescence is a particularly challenging and overwhelming period of development often characterised by confusion, and moments of ambivalence. These factors can adversely affect the uptake of interventions for adolescents and one way to address this is to create a sense of ownership of the intervention from the start. Adolescents were consulted for their views on what health conditions were important to them and that they would want to see in the Y-Check intervention. Adolescents were very clear regarding where they want to receive the check-ups (they preferred school/community) and the character (respectful, friendly) and qualifications (well-trained qualified staff) of the individuals providing the check-up.

Thirdly, the research team explicitly agreed among themselves on the criteria for selecting content of the intervention at the start of the design process, which allowed the formative research to be focused and ensured there was fidelity in the process. For example, we were clear from the start that the choice of interventions should be guided by important principles regarding screening interventions. It was necessary to explain to (and get the support of) key stakeholders that we will only conduct routine check-ups (screening) for conditions with an accurate and acceptable test for which an effective intervention is locally available and accessible to adolescents. This meant that not everything that was suggested by stakeholders was taken forward; this was carefully communicated during the informed consent process and in the consensus-building workshop at the end of this phase of the research programme.

Another key learning is the importance of thinking through issues of yield in terms of numbers available to screen and follow-ups. In our study we carefully examined data on school attendance and the transition from basic school to senior high school (SHS). Our study setting Cape Coast is particularly unusual in terms of SHS education because it has some of the best SHSs in Ghana and thus attracts many (usually affluent) students from outside the Metropolitan area of Cape Coast. This phenomenon presents some challenges in ensuring adolescents are able to be followed up after screening.

Lastly, learning from across the research teams in Ghana, Tanzania and Zimbabwe who were conducting similar formative research was extremely useful. This enabled us to appreciate the unique strengths and weaknesses in the current school health programme in Ghana and the need to explore how to leverage this infrastructure to design Y-Check Ghana. The fact that the study was coordinated by WHO enhanced the credibility of the study in the eyes of many of the key informants. This programme is of key interest to WHO and the evidence to be generated will help inform the development of guidelines related to routine health check-ups for adolescents among low- and middle-income (LMIC) countries.

Nevertheless, this study had limitations. Firstly, we had planned to conduct a series of three participatory workshops each with 20 younger and 20 older adolescents in the four study communities, but this was later revised to holding only one workshop for each group with 12

participants in each community due to COVID-19 restrictions. The reduced sample, including the fact that this study was conducted only in Cape Coast, may limit the generalizability of the opinions recorded in this study. Whilst this is an important weakness, our reliance on multiple sources of information would have lessened the effect. Secondly, we had planned to recruit parent-adolescent dyads for the separate participatory workshops with parents and adolescents to provide opportunities to clarify and cross-validate discussions during the workshops. However, this was logistically challenging because, in most instances, the parents were not immediately available to participate (though they were happy to grant consent for their children to participate). We do not think this outcome adversely affected the credibility of the workshop findings because, on the contrary, the preferences recorded are likely to be more independent and devoid of possible bias or influences.

## Conclusion

The design process has resulted in an intervention whose content and structure is determined by the local context and which builds on best practice. The design process is continuing into phase two of the programme (2022–2024) which will involve a pilot test to further refine the Y-Check intervention, followed by implementation research to rigorously evaluate the feasibility, acceptability, coverage, yield of previously undiagnosed conditions and cost of these health check-ups.

## Supporting information

**S1 Appendix.**
(PDF)

## Acknowledgments

This phase of the multi-country Y-Check Adolescent Health and Well-being Research Programme was coordinated by David Ross and Kid Kohl while they were working for the World Health Organization´s Maternal, Newborn, Child and Adolescent Health and Ageing Department, with funding support from the Fondation Botnar.

Also, we thank Pascale Pomoshchnick and Gersende Moyse of the World Health Organization for administrative support in Geneva and Valentina Baltag of the World Health Organization for technical advice. We thank the director and staff of the family health directorate of the Ghana Health Service for their support and encouragement for the Y-Check programme.

Finally, many thanks to Emmanuel Narh, Anita Adepa Gyekye, and Esther Buernorki Noah for their role in organising the participatory workshops. We also thank all the key informants from the Ghana Health Services, Ghana Education Service, and the director of Planned Parenthood Association of Ghana, Cape Coast. Our sincere gratitude to all the adolescents who participated in the study.

## Author Contributions

**Conceptualization:** Benedict Weobong, David A. Ross.

**Data curation:** Franklin N. Glozah, Hannah B. Taylor-Abdulai, Eric Koka, Nancy Addae, Stanley Alor.

**Formal analysis:** Benedict Weobong, Franklin N. Glozah, Eric Koka, Nancy Addae, Stanley Alor, Kid Kohl, Philip B. Adongo.

**Funding acquisition:** David A. Ross.

**Methodology:** Kid Kohl, Prerna Banati, Philip B. Adongo, David A. Ross.

**Resources:** Prerna Banati.

**Supervision:** Benedict Weobong.

**Writing – original draft:** Benedict Weobong.

**Writing – review & editing:** Franklin N. Glozah, Hannah B. Taylor-Abdulai, Eric Koka, Nancy Addae, Stanley Alor, Kid Kohl, Prerna Banati, Philip B. Adongo, David A. Ross.

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
