## [Decision Letter · Decision Letter 0]

19 Sep 2023

PONE-D-23-21967Reaching adolescents with health services: systematic development of an adolescent health check-ups programme in Ghana (Y-Check, Ghana)PLOS ONE

Dear Dr. Weobong,

Thank you for submitting your manuscript to PLOS ONE. After careful consideration, we feel that it has merit but does not fully meet PLOS ONE’s publication criteria as it currently stands. Therefore, we invite you to submit a revised version of the manuscript that addresses the points raised during the review process.

We look forward to receiving your revised manuscript.

Kind regards,

Yolanda Malele-Kolisa, BDS, MPH, MDent, PhD

Academic Editor

PLOS ONE

Journal Requirements:

5. We note that you have referenced (unpublished) on page 9, which has currently not yet been accepted for publication. Please remove this from your References and amend this to state in the body of your manuscript: (ie “Bewick et al. [Unpublished]”) as detailed online in our guide for authors

Reviewers' comments:

Reviewer's Responses to Questions

**Comments to the Author**

1. Is the manuscript technically sound, and do the data support the conclusions?

Reviewer #1: Yes

2. Has the statistical analysis been performed appropriately and rigorously? 

Reviewer #1: No

3. Have the authors made all data underlying the findings in their manuscript fully available?

Reviewer #1: No

4. Is the manuscript presented in an intelligible fashion and written in standard English?

Reviewer #1: Yes

5. Review Comments to the Author

Reviewer #1: It is exciting to see a manuscript focused on the improvement of adolescent health in sub-Saharan Africa. Please, find the comments below.

Methods

• This is a huge study that covered different locations in Cape coast, Ghana. For clarity, the study setting should be separated from the design. This will enhance the appreciation of the uniqueness and content of the design.

• Details about the data analysis was inadequate despite the huge data used in this research. For example, for ‘Identification of key policies, programmes, and interventions‘ step, how were the analysis of the identified policies, programs and interventions carried out?

The same applies to ‘Understanding the school health system and school attendance’ step. How were the data and literature identified reviewed and how was school attendance and access evaluated?

Results:

• The presentation of the results is not following the sequence of the steps in the design. Presenting the results in the logical flow of the steps in the design will make the reading easier. For example, the third step in the design was to identify common adolescent health problems but this was the first result to be presented. It is confusing.

• It appears that the data for identification of the adolescent health problems were not analyzed. What were presented in tables 1 and 2 appears like data from a single study/document (which was not referenced in this section where it was mentioned). The data from the literature search and the key informant interviews should be correctly analyzed and presented clearly to answer this objective. The method of analysis should also be clearly spelt out.

• The description of the study settlements, school characteristics (lines 329-332) presented should have been under study settings. They are not part of the results.

• The description of the participants in lines 282-285 is difficult to comprehend. It should be made clear.

• Please, provide a reference for the definition of ‘adolescent friendly spaces’ (lines 479-481). It appears that the only question that adolescents ask there is only about sex. This is incorrect and misleading.

• The demographic characteristics of the Community Advisory Board should be added for better understanding of their contributions.

• Lines 361-363- This is discussion and should be deleted. All other discussions (e.g. Lines 389-401, 365-366) in this section should be transferred to discussion section.

General comments

• Some sentences are not clear- lines 223-226, 282-285

• Some sentences are also too long making comprehension difficult e.g., lines 239-245

• There was a mixture of British and American English. Please, stick to one.

Tables

• What does Y mean in Table 6? This should be defined.

Mid upper arm circumference is used for screening for malnutrition in early childhood. It has no place in adolescence.

6. PLOS authors have the option to publish the peer review history of their article (what does this mean?). If published, this will include your full peer review and any attached files.

Reviewer #1: No

---

## [Author Response · Author response to Decision Letter 0]

18 Nov 2023

Thank you sincerely for your detailed review and feedback. We have now prepared a point-by-point response to each of your comments and hope these meet your expectations.

---

## [Decision Letter · Decision Letter 1]

10 Jan 2024

PONE-D-23-21967R1Reaching adolescents with health services: systematic development of an adolescent health check-ups programme in Ghana (Y-Check, Ghana)PLOS ONE

Dear Dr. Weobong,

Thank you for submitting your manuscript to PLOS ONE. After careful consideration, we feel that it has merit but does not fully meet PLOS ONE’s publication criteria as it currently stands. Therefore, we invite you to submit a revised version of the manuscript that addresses the points raised during the review process.

We look forward to receiving your revised manuscript.

Kind regards,

Yolanda Malele-Kolisa, BDS, MPH, MDent, PhD

Academic Editor

PLOS ONE

Additional Editor Comments :

The reviewers proposed significant changes to the manuscript to increase the scientific rigor of the methods. This is a standard requirement for the journal to comply with rigorous methodology.

Rev 2 suggested the following: PLEASE create a point-by-point response to the comments.

1. The methodology and results sections need major reframing particularly given that multiple forms of data were utilized.

2. On one of your diagrams, you clearly showed how the study involved three phases which included situation analysis, feasibility & acceptability assessments with participants and co-development of an intervention.

3. The methodology & results are read there is lack of coherence in terms of that illustration.

4. There are methodological gaps on who exactly the "key informants" comprised of?

5. lack of clarity on search strategy of undertaking literature review?

6. How did you analyze it?

7. What tools did you use for assessing feasibility and acceptability?

8. There is need for more details in the steps you took in your framework analysis, how did you ensure trustworthiness?

In summary, you need to be clear, what data you collected, tools for collecting that data, what information that data provided and how did you analyse it.

9. This all needs to be aligned with your three phrases.

10. Lastly your results are too detailed as you have incorporated elements of discussions in them. That needs to be ironed out.

Reviewers' comments:

Reviewer's Responses to Questions

**Comments to the Author**

1. If the authors have adequately addressed your comments raised in a previous round of review and you feel that this manuscript is now acceptable for publication, you may indicate that here to bypass the “Comments to the Author” section, enter your conflict of interest statement in the “Confidential to Editor” section, and submit your "Accept" recommendation.

Reviewer #2: (No Response)

Reviewer #3: (No Response)

2. Is the manuscript technically sound, and do the data support the conclusions?

Reviewer #2: No

Reviewer #3: Yes

3. Has the statistical analysis been performed appropriately and rigorously? 

Reviewer #2: No

Reviewer #3: Yes

4. Have the authors made all data underlying the findings in their manuscript fully available?

Reviewer #2: Yes

Reviewer #3: Yes

5. Is the manuscript presented in an intelligible fashion and written in standard English?

Reviewer #2: Yes

Reviewer #3: Yes

6. Review Comments to the Author

Reviewer #2: This is a very interesting and important study however the methodology and results sections need major reframing particularly given that multiple forms of data were utilized. On one of your diagrams you clearly showed how the study involved three phases which included; situation analysis, feasibility & acceptibility assessments with participants and co-development of an intervention. However, when the methodology & results are read there is lack of coherence in terms of that illustration.

There are methodological gaps on who exactly the "key informants" comprised of? lack of clarity on search strategy of undertaking literature review? How did you analyze it? What tools did you use for assessing feasibility and acceptability? There is need for more details in the steps you took in your framework analysis, how did you ensure trustworthiness?

In summary, you need to be clear, what data you collected, tools for collecting that data, what information that data provided and how did you analyse it. This all needs to be aligned with your three phrases.

Lastly your results are too detailed as you have incorporated elements of discussions in them. That needs to be ironed out.

Reviewer #3: The study conducted by the authors provides valuable insights into adolescent health and significantly contributes to the existing literature in the field. The methodology is robust, the results are well-supported, and the discussion is thought-provoking. The findings presented in the manuscript have the potential to make a meaningful impact and I am confident that it will be of interest to the journal's readership. I recommend this manuscript for publication without reservation.

7. PLOS authors have the option to publish the peer review history of their article (what does this mean?). If published, this will include your full peer review and any attached files.

Reviewer #2: **Yes: **Mpho Matlakale Molete

Reviewer #3: No

---

## [Author Response · Author response to Decision Letter 1]

19 Feb 2024

We thank you for your helpful comments. We have provided a rebuttal to each of these in the attached files

---

## [Editor Report · Decision Letter 2]

14 May 2024

Reaching adolescents with health services: systematic development of an adolescent health and wellbeing check-ups programme in Ghana (Y-Check, Ghana)

PONE-D-23-21967R2

Dear Dr. Benedict Woebong

We’re pleased to inform you that your manuscript has been judged scientifically suitable for publication and will be formally accepted for publication once it meets all outstanding technical requirements.

Kind regards,

Sogo France Matlala, PhD

Academic Editor

PLOS ONE
---

## [Editor Report · Acceptance letter]

12 Jun 2024

PONE-D-23-21967R2 

PLOS ONE

Dear Dr. Weobong, 

I'm pleased to inform you that your manuscript has been deemed suitable for publication in PLOS ONE. Congratulations! Your manuscript is now being handed over to our production team.

Kind regards, 

on behalf of

Professor Sogo France Matlala 

Academic Editor

PLOS ONE